# Newcastle Disease Virus Virotherapy: Unveiling Oncolytic Efficacy and Immunomodulation

**DOI:** 10.3390/biomedicines12071497

**Published:** 2024-07-05

**Authors:** Kawther A. Zaher, Jehan S. Alrahimi, Fatemah S. Basingab, Alia M. Aldahlawi

**Affiliations:** 1Immunology Unit, King Fahd Medical Research Center, King Abdulaziz University, Jeddah 21589, Saudi Arabia; 2Department of Medical Laboratory Sciences, Faculty of Applied Medical Sciences, King Abdulaziz University, Jeddah 21589, Saudi Arabia; 3Department of Biological Sciences, Faculty of Science, King Abdulaziz University, Jeddah 21589, Saudi Arabia

**Keywords:** oncolytic virus, virotherapy, virus targeting, immunotherapy, combination therapy, tumor cell lysate

## Abstract

In virotherapy, cancer cells are eradicated via viral infection, replication, and dissemination (oncolysis). Background: This study aims to evaluate the oncolytic potential of Newcastle disease virus (NDV) against colon cancer and explore the immune response associated with its therapeutic effects. Methods: NDV was tested for its oncolytic potential in colon cancer cell lines using MTT assays and apoptosis assessments. Tumor-induced mice were treated with NDV, tumor cell lysate (TCL), or a combination of both. After the euthanasia of murine subjects, an assessment of oncolytic efficacy was performed through flow cytometry analysis of murine blood and tumor tissue, targeting CD83, CD86, CD8, and CD4. An ELISA was also performed to examine interferon-gamma levels, interleukin-4 levels, interleukin-12 levels, and interleukin-10 levels in serum and spleen homogenate. Results: Cell viability was low in HCT116 and HT-29, indicating a cytotoxic effect in the MTT assay. NDV+TCL recorded the highest rate of cell death (56.72%). NDV+TCL had accelerated cell death after 48 h, reaching 58.4%. The flow cytometry analysis of the blood and tumor of mice with induced tumor treated with combined treatment revealed elevated levels of CD83, CD86, CD8, and CD4 (76.3, 66.9, 83.7, and 14.4%, respectively). The ELISA levels of IFN-γ, IL-4, and IL-12 in serum and the spleen homogenate were elevated (107.6 ± 9.25 pg/mL). In contrast, the expression of IL-10 was significantly reduced (1 ± 0.79).

## 1. Introduction

An oncolytic virus (OV) kills tumor cells through active infection and viral spread. The discharge of tumor antigens against a background of molecular patterns associated with the pathogen induces local and systemic antitumor immunity, creating an immune response to the tumor microenvironment [1,2,3,4,5,6,7,8,9,10]. Productive infection and viral dissemination destroy tumor cells directly. Recently, viral oncolysis has been emphasized in clinical oncology. Studies focus on using oncolytic viruses (OVs) for amalgamated treatment protocols that elicit antineoplastic reactions and/or induce immune cell infiltration into malignant tumors. Immunotherapy is becoming increasingly common as a treatment option in oncology, thus highlighting this emphasis. A virus uses steric and electrical properties to interact with a specific biological target and activate (or inhibit) its natural response.

NDV is an avian paramyxovirus with oncolytic properties that target human cancers [11,12,13,14,15]. Despite NDV having a sialic acid receptor in mammalian cells, virus replication is constrained. The virus can infect mammalian cells. Certain avian species are the inherent carriers of NDV [14]. According to recent reports [12,15], an attenuated strain of Newcastle disease virus (NDV) can induce a singular replication cycle in mammalian cells that have been infected. The variegated population of viruses of the NDV clone can be traced back to a solitary clone obtained from the NDV Hitchner B1 parental strain. The recent clone that has gone viral exhibits a low intracerebral pathogenicity index (ICPI), which suggests that it has undergone attenuation through multiple passages in specific-pathogen-free (SPF) eggs. This contrasts with the progenitor NDV Hitchner B1, which has an ICPI of 0.93. The observed low ICPI value suggests that the recently identified viral clone has undergone a process of attenuation. The attenuated virus has exhibited selective cytotoxicity towards cancer cells, explicitly targeting murine and human lung carcinomas. Although initially developed for use in poultry, its potential as an anti-cancer agent has been demonstrated. The oncolytic activity of the virus is founded on apoptosis, an inherent mechanism of cellular demise [16]. Newcastle disease virus (NDV) attenuated strains can induce a singular replication cycle in mammalian cells [12,15]. A solitary clone from the NDV Hitchner B1 parental strain produced the variegated population of the NDV clone. Viral clones with low intracerebral pathogenicity index (ICPI) suggest they have undergone multiple passages in specific-pathogen-free (SPF) eggs.

In contrast, NDV Hitchner B1 has an ICPI of 0.93. A low ICPI value indicates attenuation of the newly identified virus. Attenuated viruses specifically target murine and human lung carcinomas. Despite its poultry origins, it has been proven to be an anti-cancer agent. The virus’s oncolytic activity is attributed to apoptosis [16]. In response to RNA viruses such as NDV that cause infection and cytosolic replication, cells produce type I interferon (IFN). This chemical reaction reduces viral replication and dissemination. IFN-triggered factors with antiviral properties maintain the antiviral state. Research has been conducted on myxovirus resistance proteins and intrinsic antiviral enzymes [17]. NDV promotes immunogenic cell death (ICD) in cancer cells, including glioblastomas, lung cancers, and melanoma. The literature documents this [18,19,20,21]. According to Koks et al. [18], a mouse model of orthotopic immunocompetent GL261 glioma induced by oncolytic NDV primes adaptive antitumor immunity. Recent studies have shown that NDV reduced the expression and release of ICD markers in melanoma cells [20,21]. Further investigation is required to establish whether oncolytic NDV successfully triggers ICD in specific malignancies.

It has been confirmed that NDV induces an increase in tumor necrosis factor-alpha (TNF-α), interferon-gamma (IFN-γ), and interferon 1 (IFN1) [13,22,23]. Moreover, numerous studies have examined the oncolytic efficacy of NDV in mice, and none of them have documented tumor metastasis in the NDV-treated cohort. Additionally, the activation of cytotoxic T lymphocytes (CTLs) is accompanied by the enhancement of CD4^+^ T helper cells, and the collaboration of CD8^+^ and CTLs has been observed [24].

This investigation aims to ascertain whether the oncolytic Newcastle disease virus (NDV) elicits immunogenic cell death (ICD) in colon cancer cells, specifically focusing on HCT116 and HT-29 cells. Furthermore, this study assesses the oncolytic potential of NDV infection alone and/or tumor cell lysate (TCL) in mice with colon cancer. The efficacy of this treatment was evaluated through flow cytometry analysis for CD83, CD86, CD8, and CD4 markers, as well as ELISA on mouse spleen for IL-4, -10, -12, and IFN-γ. This was performed to enhance our understanding of its functionality.

## 2. Materials and Methods

### 2.1. Cells and Virus

HCT116 (human colorectal carcinoma cell line) and HT-29 cells (human colorectal adenocarcinoma cell line) were sourced from the Immunology Unit of King Fahd Medical Research Center (KFMRC), King Abdulaziz University (KAU), Jeddah, Saudi Arabia. At the same time, SL-27 (embryo-fibroblast cells, CRL-1590™) were purchased from ATCC^®^. DMEM (ThermoFisher, Waltham, MA, USA) was utilized to grow the cells at 37 °C in an incubator with 5% CO_2_, 1 mM of L-glutamine/mL, 10% heat-inactivated fetal bovine serum (Gibco, Billings, MT, USA), and a 1% penicillin/streptomycin solution with 10,000 units and 10,000 g, respectively (HyClone, South Logan, UT, USA) improved the media. The Hitchner B1 strain of Newcastle disease virus employed in this study, provided by Professor Zaher, is a mild strain owing to 22 passages on embryonated chicken eggs. 

### 2.2. Virus Titration in SL-27 Cells

The virus was titrated using a plaque assay [25]. Briefly, the chicken-embryo fibroblast cells (SL-27) were grown in 6-well tissue culture dishes with 5 × 10^5^ cells per well until confluency. The cells were infected with a 10-fold serial dilution of NDV in triplicate wells with 1 mL of the 10^−6^, 10^−7^, and 10^−8^ viral dilutions for 2 h. After aspirating the viral inoculum, each well was covered with 2 mL of complete MEM with methylcellulose and incubated for two days with daily inverted microscope observation until cytopathic effects (CPE) like rounding developed. After removing the medium, each well received 0.5 mL of 0.1% crystal violet. Incubation lasted 20 min. Washing and drying the dishes removed the discoloration. Counting plaques in each well and multiplying by the dilution factor gave the titer.

### 2.3. Preparation of Tumor Cell Lysate

The generation of tumor cell lysates (TCL) was conducted using the method described [26]. In brief, the HCT116 and HT-29 tumor cell pellets were re-suspended in phosphate-buffered saline (PBS) to 1 × 10^7^ cell/mL, ice-cold, followed by five cycles of freezing in liquid nitrogen (for 5 min) and thawing in a warm water bath at 37 °C (for 5 min). The lysates were centrifuged for ten minutes at a speed of 2000× *g* to segregate the cellular detritus. The BCA test was used to determine the protein concentration, where the protein solution was then diluted to the desired concentration for both in vivo and in vitro studies. The BCA was performed using the PierceTM Rapid Gold BCA Protein Assay Kit (Catalog number: A53225, Pierce Biotechnology, Inc., ThermoFisher Scientific, 3701 N. Meridian Road, Rockford, IL, USA). 

### 2.4. In Silico Oncolytic Activity of NDV on HCT-116 and HT-29 Cells

#### 2.4.1. MTT Assay

Per manufacturer instructions, cell viability was determined using an MTT assay [27], Kit (#CGD1-1KT, Sigma-Aldrich, St. Louis, MO, USA). HCT116 and HT-29 tumor cells (4 × 10^4^ cells/mL) were grown in DMEM with 10% FBS for three hours. After adding 100 μL of this cell suspension to each well of a 96-well microtiter plate, the plate was put in an incubator at 37 °C and 5% CO_2_ overnight to enable cell adherence before any treatments were applied. After 24 h, fresh DMEM with 20 μL serial dilutions of each treatment was introduced. The three treatments were the virus alone, tumor cell lysate, and both. Duplicate controls of untreated cells were created. The cells returned to the incubator for 24 h. Then, 120 μL was carefully pipetted, and 100 μL of 10% MTT solution in the medium in each plate was added per well; three to four hours of incubation followed. After incubation, the purple formazan crystals were dissolved by removing the culture medium and adding an equal amount of culture media and 100 μL of MTT solvent into each well. After that, the plate was placed on a plate shaker for thirty minutes at 120 rpm until a consistent color was discovered, indicating that all crystals had dissolved. Finally, a spectrophotometer at 570 nm calculated the crystals’ concentration. 

#### 2.4.2. Cell Apoptosis Evaluation Using an Annexin V Kit

The proportion of cells that underwent apoptosis was estimated by counting the number of positive cells for Annexin V and negative for PI [19,21]. The cells were distributed at a density of 1 × 10^5^ cells per well in 12-well plates for the trypan blue exclusion experiment. The cells were then infected with 10 PFU of NDV for 12, 24, and 48 h with or without the virus. Following staining with 0.4% trypan blue, the number of cells was determined through microscopic examination. For the staining procedure, a 0.4% solution of trypan blue was used to treat the cells. Dual labeling with fluorescein isothiocyanate (FITC)-conjugated Annexin-V (BD Bioscience #C3-4554; Annexin V-FITC, Franklin Lakes, NJ, USA) and propidium iodide (PI) staining was utilized to determine the degree of cellular death, and analysis was performed using flow cytometry (Beckman Colter Life Science, Brea, CA, USA). This strategy was used because it is described in greater detail in other sources. In addition, quantitative analysis was performed on the cellular population corresponding to apoptotic cells in the lower right quadrant, which were negative for PI and positive for Annexin V [20].

### 2.5. In Vivo Protocol

#### 2.5.1. Mice

A cohort of BALB/c mice, with a mean weight of 19.58 ± 0.19 g and an age range of 6–8 weeks, were procured from the Animal House and bred under specific pathogen-free conditions at KFMRC. The mice were housed in-house and provided ad libitum water access and a nutritionally balanced diet. The specimens were maintained in controlled laboratory settings, with a temperature of 22 °C (±2), humidity levels ranging from 40–60%, and a 12 h light-dark cycle. The cages utilized were equipped with wire bottoms. Mice were weighed every five days.

#### 2.5.2. Ethical Approval

The Animal Care and Use Committee (ACUC) at KFMRC, KAU, reference number 07-CEGMAR-BIOETH 2022, approved all of the studies conducted on animals. 

#### 2.5.3. Experimental Design 

##### Lethal Toxic Dose-50 (LTD50) 

Four groups of mice were given serial 2-fold dilutions of the oncolytic virus (start at 3 × 10^7^) suspended in 0.2 mL TCL containing 4 μg protein (injected I/M in the right flank), while the fifth group was left as a control. The mice were observed daily for ten days. 

##### Animal Inoculation 

Experiment 1: Assessment of Oncolytic Activity

BALB/c female mice were purchased from KFMRC’s Animal House. Mice were anesthetized by Isoflurane inhalation. An incision of around 1 cm was made in the lower abdomen, and the colorectum was pulled out. Under a stereoscopic microscope, 1.5 × 10^7^ tumor cells from xenograft mice were obtained. The cells were suspended in 0.2 mL saline and injected into the colorectum subserosal layer using a 30 G bent needle, not hurting vessels [27]. The wounds were closed using an AutoClip applier (FST). The Animal Care and Use Committee (ACUC) approved all animal experimental protocols at KFMRC, KAU. The tumor cells were injected in four groups (n = 5), while the fifth was injected with saline and left as a negative control. After 12 weeks, the treatment was injected intramuscularly. The first group was injected with saline to be a positive control, while groups 2–4 each received a different treatment. For instance, the second group received 0.2 mL of NDV 3 × 10^7^, the third group received 0.2 mL of TCL, and the fourth group received 0.2 mL of 3 × 10^7^ NDV with TCL. The fifth group was the negative control. The treatments were repeated on the 15th and 30th day. Mice were weighed every five days. The syngeneic mice were euthanized humanely on the 40th day, and tumor tissues were collected. A 4% paraformaldehyde solution was used overnight to fix tissues embedded in paraffin blocks the next day. Four μm sections were made and stained with hematoxylin and eosin (H&E). Images were taken using a light microscope (Zeiss, Carl Zeiss Microscopy GmbH, Jena, Germany) operated by ZEN software, volvox25_rotati.czi (2020).

Experiment 2: Assessment of Immunomodulation

Tumors were induced in BALB/c mice in four groups chemically with 15 µg of 1,2-dimethylhydrazine (DMH)/gm according to [28]. According to [29,30], different tumors formed within 14–16 weeks [28]. The fifth group (5 mice) was left as a negative control and received saline at each due injection. After that, the first group was considered to be control positive; the tumor was induced and received saline treatment. The cancer was caused in groups 2–5. Still, each group (2–4) received different treatment as in the previous experiment; the blood, lymph nodes, and spleens were collected and utilized to evaluate dendritic cell maturation and T cell activation through flow cytometric analysis. Monoclonal antibodies were used for each marker. For dendritic cell (DC) markers: anti-mouse CD80 monoclonal antibody FITC, BioLegend (BioLegend^®^, San Diego, CA, USA, Catalog #104705), anti-mouse CD86 monoclonal antibody APC, eBioscience‚™ (eBioscience™, 10255 Science Center Dr, San Diego, CA, USA, Cat #17-0862-82), and anti-mouse CD11c monoclonal antibody PE, Thermo Fisher (Thermo Fisher Scientific Inc., Waltham, MA, USA, Catalog #MHCD11C04). T cell markers used were anti-mouse TCR γδ monoclonal antibody PE, BioLegend (BioLegend, San Diego, CA, USA, Cat #118107), anti-mouse CD3 monoclonal antibody FITC, abcam (abcam™, Waltham, MA, USA, ab34722), anti-mouse CD4 monoclonal antibody eFluor™ 450, eBioscience™ (Thermo Fisher Scientific Inc., Catalog #13-0041-82), anti-mouse CD8 monoclonal antibody Alexa, eBioscience™ (Thermo Fisher Scientific Inc., Catalog #56-0081-82). The collected cells were stained and analyzed through flow cytometry (Beckman Colter Life Science, USA) using 10,000 events to detect DC maturation and T cell subpopulations. Data were then analyzed using FlowJo^TM^ version 10 software (Becton Dickinson, Franklin Lakes, NJ, USA). The mice serum and spleen homogenate were used to detect IFN-γ, IL-4, -10, and -12 levels using ELISA kits. The kits used for ELISA were as follows: Mouse interferon gamma ELISA kit (abcam™, Cambridge, UK, ab282874), Mouse IL-4 ELISA kit (abcam™, Cambridge, UK, ab100710), Mouse IL-10 ELISA kit (abcam™, Cambridge, UK, ab255729) and IL-12 Mouse ELISA kit (Thermo Fisher Scientific Inc., USA, Catalog #BMS616). 

### 2.6. Statistical Analysis

The statistical analyses were conducted utilizing the student’s *t*-test using Microsoft Excel software (Microsoft, Redmond, WA, USA). The outcomes were presented as the mean ± SD of a minimum of three autonomous trials. To evaluate the in vivo oncolytic effects, statistical significance between groups was determined using the LSD test and one-way ANOVA test. Statistical significance was attributed to differences with a *p*-value of less than 0.05.

## 3. Results

The obtained virus was propagated on SL-27 cells and titrated. A plaque assay was performed to measure the concentration of the infectious virus that makes plaques using a methylcellulose layer to fix the cell and crystal violet to visualize the dead cell area due to the virus’s cytopathic effect. The multiplicity of infection (MOI) was calculated by dividing the number of virus particles added to a well by the number of cells seeded in this well. 

### 3.1. Direct Effect of NDV on Colon Cancer Cells

The virus showed rounding of HCT-116 and HT-29 cells (Figure 1) 24 h after infection. The cells lose their standard structure, and several cytoplasmic degradations occur. The virus propagates in the tumor cell, showing almost the same CPE; syncytial formation appears 24 h after infection (Figure 1). 

### 3.2. Estimated Oncolytic Activity of NDV by MTT Assay

The viability of HCT116 and HT-29 cells was evaluated using the MTT assay, which involves 3-(4,5-dimethylthiazol-2-yl)-2,5-diphenyltetrazolium bromide. The MTT assay’s tetrahydrozoline agent can estimate the NADPH-dependent intracellular oxidoreductase activity. The evaluation of mitochondrial enzyme activity relies on reducing a tetrazole yellow dye to form purple formazan with a crystalline morphology in viable cells. Cell proliferation absorbance was quantified through an ELISA plate reader at a wavelength of 570 nm.

The lytic capacity of NDV was examined by utilizing the MTT assay. Furthermore, the individual administration of each treatment was conducted to assess the efficacy of each therapy in isolation (refer to Figure 2). The transformation of MTT dye into a purple formazan that was not soluble in water was attributed to metabolically active cells, specifically the mitochondria. Distinct variations in cell densities were observed under the microscope in both cells subjected to varying concentrations of NDV and TCL compared to the remaining treatments. Conversely, elevated concentration levels exhibit a significantly harmful impact on cellular viability. Cell viability was determined through three separate experiments, and a notable variance was observed among concentrations (as determined by the student’s *t*-test using Microsoft Excel software with a significance level of *p* < 0.001). A decline in cell viability was observed in correlation with increasing concentrations. The control group exhibited significant differences compared to all concentrations, except for the lower concentrations of 3 × 10^7^ PFU of NDV with 4 μg/mL TCL. This particular concentration was deemed marginally significant (*p* = 0.049) and (*p* = 0.025) for HCT116 and HT-29 cells, respectively, suggesting that there is no genuine distinction between lower concentrations of the treatments concerning the viability of both, as depicted in Figure 2. 

### 3.3. Oncolytic NDV Induces Apoptosis in Colon Cancer Cells

Apoptosis was analyzed by flow cytometry with FITC-conjugated Annexin V and PI double staining. NDV infection significantly increased apoptotic cell numbers compared to controls at 24 h post treatment (hpt) (Figure 3). The highest rate of cell death recorded by NDV+TCL was approximately 56.72%, followed by TCL and NDV-treated wells (50.1%, and 39.8%, respectively). Cell death rates accelerated for NDV+TCL, TCL, and NDV-treated wells after 48 hpt, reaching 58.4%, 53.3%, and 42.7%, respectively. Moreover, these rates continued to increase to 75.6%, 69.2%, and 77.9% for the same groups, respectively, after 72 hpt. In a colon cancer cell line, an oncolytic NDV strain induces ICD. Cell growth was substantially inhibited by NDV, TCL, and NDV+TCL infection at 12 h, 24 h, and 48 h after infection (hpt) (Figure 3A,B). 

Also, data was calculated regarding cell death rate, early apoptosis, late apoptosis, and PI values, which represent necrosis of cells for control negative, tumor cell lysate, NDV, and both NDV with TCL wells. Data taken from flow cytometry revealed significant data (*p* = 0.0039, 0.0027, 0.0045, respectively), as depicted in Figure 3B. In addition, the high percentage of early apoptosis (75.7%) adequately noticed the oncolytic effect of NDV with TCL.

### 3.4. In Vivo Mice Model Assessment

Female white BALB/c mice that ranged in age from 6–8 weeks, with a mean weight of 19.58 ± 0.19 g, were used after obtaining the acceptance of the ethical committee for all experimental procedures. NDV and TCL were tested for their lethal effect on mice and observed for ten days. None of the tested mice died after submitting to the tested dose or dilution. The highest concentration was 3 × 10^7^ for NDV (MOI = 10) suspended in 0.2 mL of TCL (containing 4 μg protein).

#### 3.4.1. Tumor Induction and Treatment Effect on Body Weight, Tumor Size, and Volume

Mice’s body weight was measured every week; there was no actual change among treatment groups regarding body weight (Figure 4A). However, the body weights of the negative control mice increased over time compared to the positive control group. The latter showed the lowest body weight by the end of the experiment (i.e., 19.8 g), and the NDV+TCL, together with NDV alone treated groups, hit the highest weight at 20.3 and 20.14 g. At the same time, the body weight of the TCL-treated group was approximately 19.79 g (means of repeated measures, *p* = 0.369).

After euthanasia, the tumor weight was measured for the positive control; the tumor was the highest (1.42 g) compared to the treatment groups NDV, TCL, and NDV+TCL (1.17 g, 0.94 g, and 0.72 g, respectively). The result was found to be significant (repeated measures ANOVA, *p* = 0.047) (Figure 4B).

#### 3.4.2. Pathological Sections

The colorectal region from syngenic mice received 1.5 × 10^7^ tumor cells from xenograft mice. The tumor cells were injected in four groups (n = 5), while the fifth was injected with saline and left as a negative control. After euthanasia, the colorectal region was removed (Figure 5A) and sent for histological sectioning. Histological sections showed decreased intensity of thickened walls, pyknotic nuclei, and ruptured cells in the colorectal region in NDV with TCL-treated mice (Figure 5B). The colon wall regained its average thickness and villi appearance, while the submucosa still showed a degree of hyperplasia (Figure 5C). Figure 5D shows the histological structure of positive control mice where the mucosa’s massive tumor mass (hyperplasia) invasion extends to the smooth muscle layer, revealing the complete picture of colorectal carcinoma. Liver metastasis occurred in the positive control groups (Figure 5E,F).

#### 3.4.3. Assessment of Treatment on Tumor-Induced Mice Using Flow Cytometry 

Heparinized blood, lymph nodes, and spleen were collected from mice after euthanasia, stained with anti-mouse monoclonal antibodies, and submitted for flow cytometry analysis to analyze the phenotypic profile of T cells and DCs. 

For T cell evaluation, helper and cytotoxic T cells were assessed using a positive gate percent value for surface molecules CD3^+^ CD4^+^ and CD3^+^ CD8^+^. The figure shows CD3^+^ CD4^+^ is highly expressed in the NDV+TCL treated group (80.1%) compared to the negative control group, with a significant value of *p* = 0.009. The NDV and TCL-treated groups reached 75.2% and 70.3%, *p* = 0.004 and 0.0008, respectively (Figure 6A). 

On the other hand, the T cell receptor (TCR), an antigen-specific receptor found on T cells (helper and cytotoxic T cells), and its presence as TCR chain γδ was assessed using gate percentage. TCRγδ, when gated with CD3^+^ CD4^+^, showed higher expression levels of negative control than positive control (9.4%, 9%, respectively). At the same time, treatment groups NDV and TCL NDV with TCL gave much lower significant levels (4.4%, 4.5%, 4%, respectively), where *p* = 0.0001, 0.0004, and 0.0009, respectively (Figure 6B). While TCRγδ gated with CD3^+^, CD8^+^ had the highest expression in the NDV+TCL group (74.3%), followed by NDV and then TCL-treated groups (70.2% and 61.7%, respectively). Meanwhile, the most negligible results with positive controls and negative controls reached 10% and 9.2%, respectively, with *p* = 0.0008, 0.0007, and 0.0009 (Figure 5C).

Myeloid DCs CD11c and CD11b are used for DC evaluation. They showed high statistical values in tumor-positive controls (17.3%) (Figure 7A). CD80 and CD86 usually assess the maturity of DCs. Gating with CD11c+ and CD11b^+^, CD80 was statistically highly expressed in the NDV with TCL treated group (75.2%), *p* = 0.00075 (Figure 7B). Furthermore, CD86+ was statistically highly expressed in the same group with the same gating (58.1%) (Figure 7C).

#### 3.4.4. ELISA Assessment of Serum and Spleen Tissue Homogenate 

After scarification, serum and the spleen were taken from mice, and both serum and the spleen homogenate were subjected to ELISA analysis. For IFN-γ, the highest concentration was in the spleen of NDV+TCL treatment (107.6 pg/mL ± 9.25), while, as expected, the TCL-treated group’s spleen IFN-γ concentration was higher (88.5 pg/mL ± 8.91) than the NDV treated group (73.9 pg/mL ± 8.7). On the other hand, the opposing group showed a higher level of IFN-γ than the positive control (63.7 ± 7.67 and 51.2 ± 9.42 pg/mL, respectively) (Figure 6A). 

When referring to IL-10, there was higher expression in the positive control (500 pg/mL ± 95.23) than in the negative control (3.7 pg/mL ± 92.91). For treated groups, it was remarkably down-regulated. Both IL-4 and IL-12 were highly expressed in treated groups, as shown in Figure 8C,D.

For IL-4, the mean negative control level was 3.59 pg/mL ± 2.97, while the mean positive tumor was 7.53 pg/mL ± 3.12. High IL-4 expression was observed in treated groups NDV, TCL, and NDV with TCL: 10.46 ± 3.7, 11.95 ± 3.89, and 20.21 ± 3.89 pg/mL, respectively. 

For IL-12, the negative control’s mean (5.72 ± 2.90 pg/mL) was higher than that of the positive control (4.21 ± 1.90 pg/mL). The highest expression was observed in the means of the treated groups with NDV, TCL, and NDV with TCL (8.11 ± 1.91, 10.20 ± 1.83, and 13.5 ± 0.89 pg/mL, respectively).

## 4. Discussion

The present investigation involved the propagation of NDV on HCT116 and HT-29 cells, two of the colon cancer cell lines. TCL was used to induce a more integrated immune response, as TCL is suitable for delivering various antigens associated with MHC class I and II molecules [26,31,32,33,34,35,36,37,38,39,40,41]. The oncolytic activity of NDV was demonstrated on cells, exhibiting various cytopathic effects (CPE) that commenced at 7 h post-treatment. As reported by Rodriguez et al., these effects included cell rounding, syncytial formation, and characteristic lesions of all NDV CPE. This comes in agreement with [42]

Conversely, the impact of the three treatments on HCT116 cells was accurately demonstrated through the late apoptosis percentage, which is a reliable indicator of the oncolytic effect on cells, particularly when the necrosis percentage was also calculated, as evidenced by the Annexin V/PI-induced apoptosis. The Annexin-V/PI staining technique is widely utilized [43] to evaluate apoptosis quantitatively. The current investigation employed the previously mentioned assay to assess the influence of NDV on the initiation of apoptosis in the infected cells. The results suggest a notable rise in the count of apoptotic and necrotic cells within the cancerous cells that underwent treatment with the three-group intervention in early apoptosis. Najmuddin et al. [44] have reported comparable findings concerning the impact of NDV on cancer cell lines. Prior studies have demonstrated that the Newcastle Disease Virus (NDV) can initiate cellular apoptosis via diverse pathways. Several phenomena were mentioned, including endocytic targeting of Rac1 GTPase in human tumorigenic cells that have undergone Ras transformation, the switch to viral protein synthesis from cellular protein synthesis, and the initiation of autophagy mediated by viral nucleoprotein (NP). In addition, viral RNA polymerase, which consists of a large protein (L) and is associated with phosphoprotein (P), facilitates virus replication. In addition, the membrane branching of the virus progeny, which is enabled by matrix protein (M) and fusion protein (F), contributes to the dissemination of Newcastle disease virus (NDV) in tumors. Finally, oncolysis is induced via apoptosis, necroptosis, pyroptosis, or ferroptosis, linked to immunogenic cell death assessed by Annexin V [45].

An in vivo study was conducted on female white BALB/c mice aged 6 to 8 weeks. Histological sections revealed massive colorectal cancer and hyperplasia of mucosal and submucosal tissue extending to the smooth muscular region in late-stage metastasis of carcinoma to liver tissue in the positive control group, while in the treated groups, the hyperplasia was much decreased, and the treatment prevented tumor metastasis. Additionally, there was a substantial reduction in tumor weights. These findings are explained by the oncolytic virotherapy of NDV, which is consistent with the research conducted by Wang et al. [46,47]. 

In evaluating the immune response, another experiment was performed where blood samples, lymph nodes, and spleen were subjected to cytometric analysis utilizing monoclonal antibodies targeting CD80, CD86, CD3, CD4, CD8, CD11b, CD11c, and TCRγδ. The presence of the significant markers CD80 and 86 characterizes the surface of fully mature dendritic cells (DCs), the maestro responsible for initiating the immune response and stimulating T and B lymphocytes [48]. CD86 is a molecule with pro-stimulatory properties expressed on the surface of dendritic cells, providing a co-stimulatory effect to T lymphocytes [49]. The upregulation of CD8^+^ results in an augmented cytotoxic effect against neoplastic cells, which were high in the treated group with NDV and TCL and the highest level in the NDV and TCL treated group. In mature DCs, MHC-peptide complexes induce conjugate stabilization, the formation of mature immune synapses, and the effective proliferation of T cells. The mature DC forms high-avidity stable conjugates and mature immune synapses and activates T cells effectively. In contrast, the immature DC establishes multiple short, low-affinity contacts and does not cluster TCRs efficiently [50]. Conversely, CD4^+^ cells differentiate into T-helper cells upon binding with MHC-II molecules, activating macrophages and B lymphocytes. It is a biomarker that is commonly used to identify and quantify T-helper cells in the immune system. The CD4 molecule is a type of transmembrane glycoprotein with a molecular weight of 55 kDa. It is a solitary polypeptide chain on the exterior of T cells restricted by MHC class II, as reported by Miceli and Parnes [49]. As a result of the increased TCRγδ in the CD3^+^-CD8^+^ group, T lymphocytes are capable of producing abundant proinflammatory cytokines such as IFN-γ, powerful cytotoxic effector functions, and MHC-independent recognition of antigens, making them an essential component of cancer immunotherapy [49]. Mittelbrunn et al. [50,51,52] confirm our findings. According to them, T cells can kill various tumor cell lines and in vitro tumors, including leukemia, neuroblastoma, and multiple carcinomas.

The ELISA test results indicate a significant elevation in IFN-γ levels, a typical finding in a viral infection. The expression of IFN-γ is notably elevated in cytotoxic T lymphocytes (CTLs), and its primary role is to augment the presentation of antigens while also regulating the balance between Th1 and Th2 cells, as reported by Tau et al. [53]. The cytokine interleukin-4 (IL-4) has been observed to possess anti-inflammatory properties. It has been noted to act on macrophages, a cell type that plays a crucial role in the inflammatory response. IL-4 has been found to induce an alternative macrophage activation pathway through its actions. The ELISA assay outcomes demonstrate elevated levels of IL-4 expression in all treated groups and the positive control group. It is possible to attribute that in mice, this cytokine serves as a protective agent against the high inflammatory effects induced by the injection of NDV. This may be considered a counter-attack defense mechanism against allergic inflammation. The elevated expression observed in the positive control group may be attributed to the influence of the tumor microenvironment, which is known to suppress inflammation and promote tumor proliferation [54]. Interleukin-10 (IL-10) possesses significant anti-inflammatory properties. It is known to have a crucial function in restricting the host’s immune response to pathogens and tumors [55].

The cytokine IL-10 has been observed to inhibit various activities of natural killer (NK) cells and T cells. This is mainly achieved by impeding the production of pro-inflammatory cytokines, such as IL-12, by antigen-presenting cells (APCs) and hindering the up-regulation of molecules involved in antigen presentation and lymphocyte activation. The ELISA findings indicate a marked increase in the expression of IL-10 in the positive control group, while the treated groups exhibited a substantial decrease in expression. The cytokine interleukin-12 (IL-12) is a pro-inflammatory heterodimer that plays a crucial role in regulating the responses of T-cells and natural killer cells. It also stimulates the production of IFN-γ and promotes the differentiation of T helper 1 (TH1) cells. Additionally, IL-12 is a vital connection between innate resistance and adaptive immunity. The findings indicate a significant upregulation of IL-12 in the treated cohorts, consistent with the observed heightened IFN-γ expression reported by Bashyam [56].

## 5. Conclusions

The present study aims to examine the oncolytic impact of the vaccinal strain of Newcastle disease virus (Hitchner B1) on the HCT116 and HT-29 colon cancer cell lines. The potent antitumor activity of Newcastle disease virus (NDV) was noted at 7, 24, and 72 h post-treatment. Additionally, confirmation was obtained through an MTT assay and apoptosis assessment utilizing Annexin V/PI staining. The oncolytic effect of NDV was observed in vivo, significantly reducing tumor size and weight in the treated groups. The upregulation of CD83 indicates the proliferation and differentiation of dendritic cells and the high expression of the proinflammatory molecule CD86. In addition, the treated groups exhibited significant expression of IFN-γ and IL-12, which stimulated the immune response to the tumor and promoted the proliferation of CTL and Th cells, ultimately leading to stimulated programmed cell death. It has been confirmed that the conjugation of NDV with TCL has yielded promising outcomes for the development of tumor vaccines in the future.

## Figures and Tables

**Figure 1 biomedicines-12-01497-f001:**
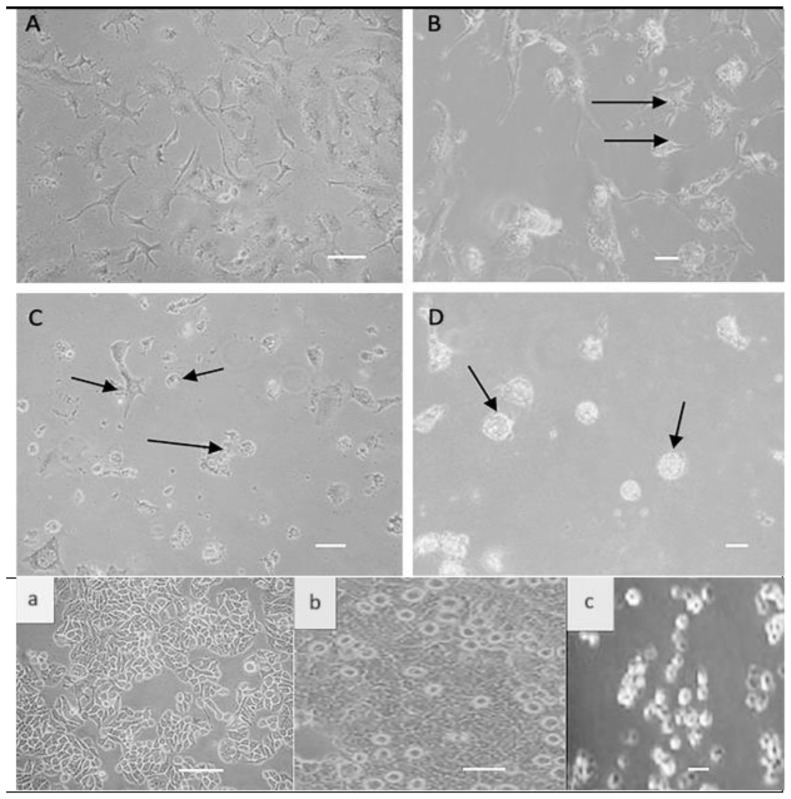
Photos of the upper two rows of HCT116 cells before and after treatment under the inverted microscope (**A**): Normal HCT-116 cells. (**B**): Cells 7 h post-infection, the cells lost their texture and shape. (**C**): Cells 24 h post-infection cell rounding and cell death. (**D**): Severe cell apoptosis, degradation, and syncytial cells are apparent. Lower row photos: HT-29 cells before and after treatment under the inverted microscope (**a**): Normal cell line. (**b**): 24 h post-infection shows cell rounding and multiple cell deaths. (**c**): Severe cell apoptosis and degradation (magnification is 40×, scale bar is 40 μm), Immunology unit, KFMRC, King Abdulaziz University.

**Figure 2 biomedicines-12-01497-f002:**
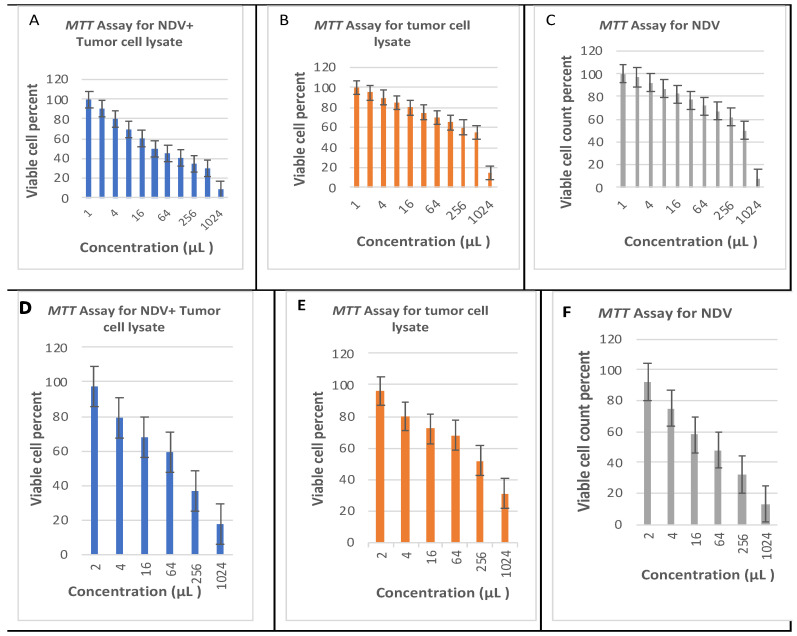
HCT116 tumor cell viability by MTT assay with different treatments from (**A**–**C**). HT-29 tumor cell viability by MTT assay with different treatments from (**D**–**F**). The bar plot represents the total percentages of cell viability from three independent experiments. The results are presented as the mean ± SD of three independent experiments.

**Figure 3 biomedicines-12-01497-f003:**
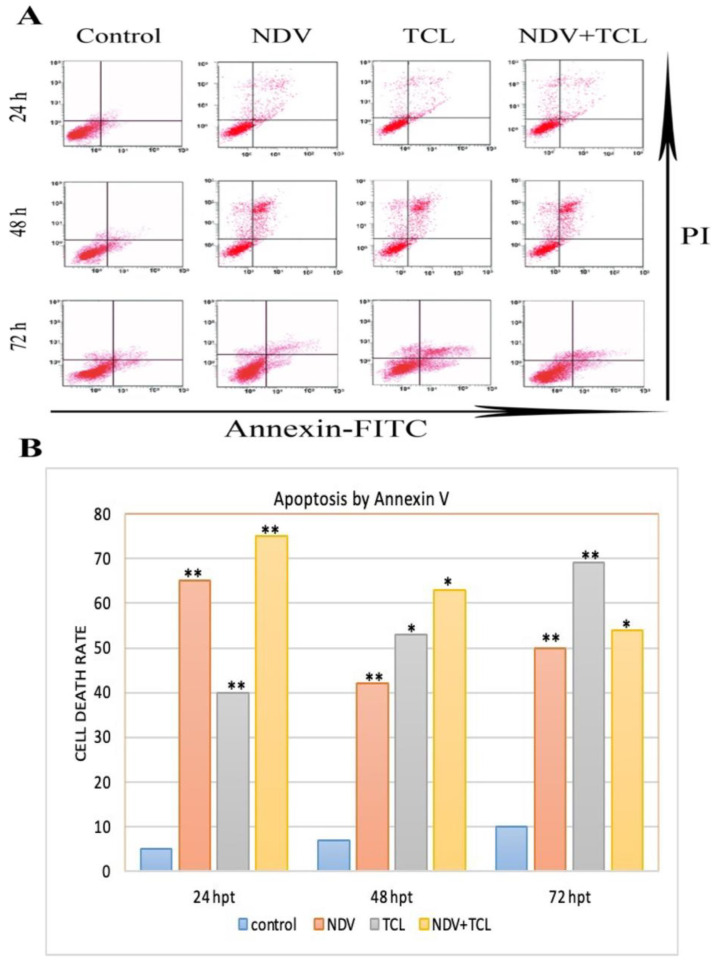
NDV apoptotic effect immunogenic cell death in colon cancer cells. (**A**) 10 PFU of NDV infected HCT116 cells in series one and with 4 μg/mL TCL in series two and both in series 3. (**B**) The results are 24, 48, and 72 hpt. Values are expressed as mean ± SEM. (n = 3, * *p* < 0.05, ** *p* < 0.005).

**Figure 4 biomedicines-12-01497-f004:**
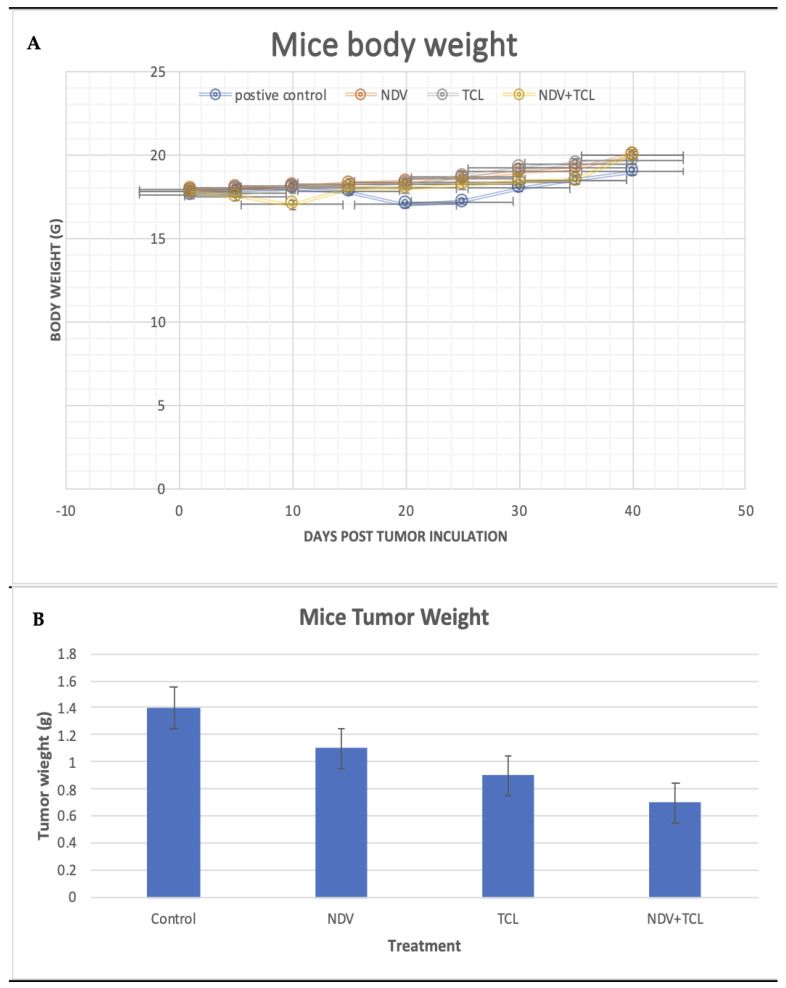
Oncolytic effects of the three treatments on tumor-induced mice (**A**) Dot plot of daily mice body weight after treatment. (**B**) Mice tumor weight after euthanasia. Values are expressed as mean ± SEM (n = 3) (repeated measures *p* = 0.047, n = 5).

**Figure 5 biomedicines-12-01497-f005:**
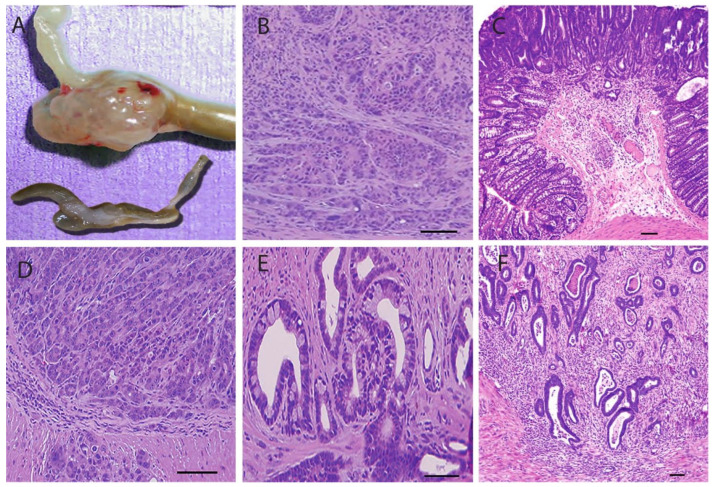
Photos were taken from mice that received 1.5 × 10^7^ tumor cells from xenograft mice. (**A**): Shows the colorectum region of the positive control and NDV-treated group. (**B**): Histological section of colorectum region of NDV treated group. (**C**): Histological section of the colon of the TCL-treated group. (**D**): Histological section reveals carcinoma in positive control mice. (**E**,**F**): Histological section showing metastasis to the liver in positive control mice. The scale bar is 20 m, and the original magnification is ×40.

**Figure 6 biomedicines-12-01497-f006:**
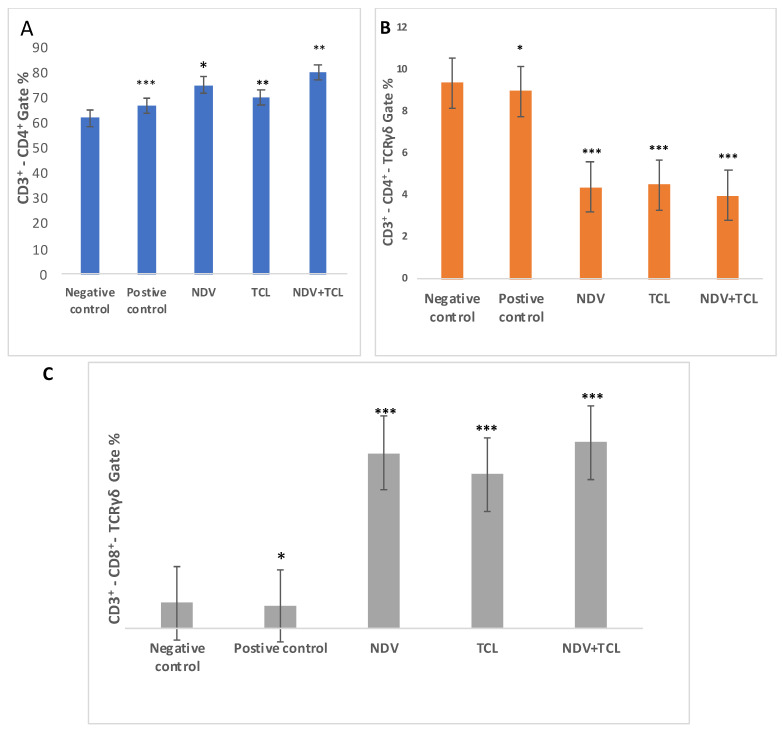
Flow cytometric analysis of the five mouse groups’ BMCs for the expression of CD3^+^-CD4^+^ (**A**), TCRγδ with CD3^+^-CD4^+^ Gate (**B**), TCRγδ with CD3^+^-CD8^+^ Gate (**C**). Data were analyzed using a one-way ANOVA test (n = 5, * *p* > 0.05, ** *p* > 0.01, and *** *p* > 0.001, where the negative-control group is compared with the rest of the groups).

**Figure 7 biomedicines-12-01497-f007:**
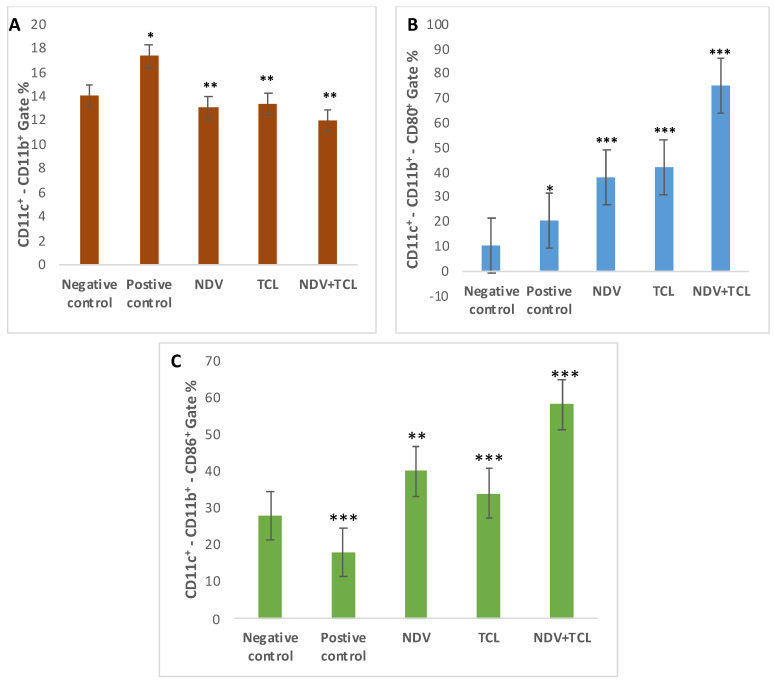
Flow cytometric analysis of the five mouse groups’ BMCs for the expression of surface molecules on dendritic cells: CD11c CD11b chart (**A**), CD11c^+^- CD11b^+^- CD80+ Gate chart (**B**), CD11c^+^- CD11b^+^- CD86^+^ Gate chart (**C**). Data were analyzed using a one-way ANOVA test (n = 5, * *p* > 0.05, ** *p* > 0.01, and *** *p* > 0.001, where the negative-control group is compared with the rest of the groups).

**Figure 8 biomedicines-12-01497-f008:**
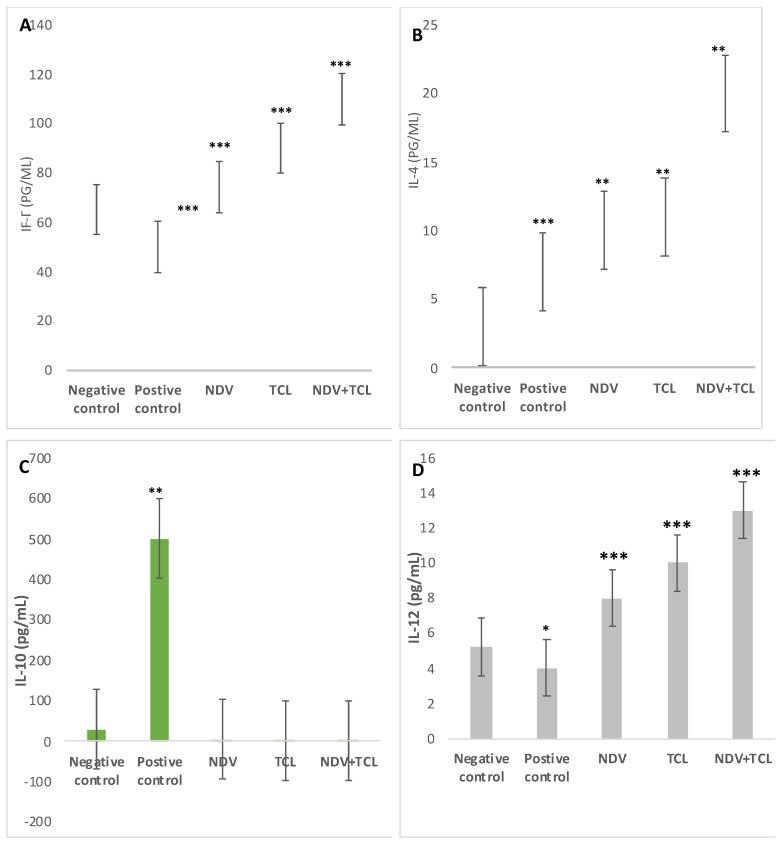
ELISA assessment of spleen homogenate to detect different cytokine levels: IFN-γ (**A**), IL-4 (**B**), IL-10 (**C**), and IL-12 (**D**). The results were analyzed using the one-way ANOVA test (n = 5, * *p* > 0.05, ** *p* > 0.01, and *** *p* > 0.001, where the negative-control group is compared with the rest of the groups). The differences were significant, where * *p* < 0.05, ** *p* < 0.0006, and *** *p* < 0.0001.

## Data Availability

Data are not available elsewhere.

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
