# Peer review of "Newcastle Disease Virus Virotherapy: Unveiling Oncolytic Efficacy and Immunomodulation"

_biomedicines, 2024, doi:10.3390/biomedicines12071497_

Round 1

Reviewer 1 Report

Comments and Suggestions for Authors

In this study by   Zaher and coworkers, the oncolytic potential of Newcastle disease virus (NDV) has been evaluated against against colon cancer. Moreover, authors  have analysed the  immune responseelicited by NDV treatment  in  mice bearing induced tumours. Authors have used  to this aim   NDV, tumour cell  lysate (TCL), or a combination of both.

 By  flow cytometry of murine blood and tumour tissue,  CD83, CD86, CD8, and CD4 levels were assessed. Interferon-gamma levels, Interleukin-4 levels, Interleukin-12 levels, were also elevated in NDV and NDV-TCL treated mice 

Although the effects of different strains of NDV against colon cancer models were already described , the study is potentially interesting however I have  some criticisms

The introduction is too long and it is not focused, , i.e. in lanes 74-80 the effects  of STAT3 inhibition are described, however these previous data  are not relevant for the study .

Authors describe the strain  NDV Hitchener B1 , but it is not clear whether this strain was used in the study

The authors have used throughout the study TCL , however it is not described the purpose of using TCL.

Authors have assessed the effects of NDV against human colon cell lines HCT16 and HT-29. However the effects have been evaluated in vivo against a mouse tumour model. I think authors should evaluate the effects of NDV also against mouse carcinoma colon cells.

Finally the discussion is too long and contains , again,  a description of the experiments performed,  instead of discussing the  results obtained. Authors must re-write d the discussion  ,  comparing the data obtained with previously published papers and clearly presenting  the limitations of the study as well the  relevance of their data.

Author Response

First of all, thank you for your time and effort in reviewing our manuscript and for your valuable remarks that helped the manuscript to be in the best form. We sincerely thank the reviewers for their valuable comments, which we will do our best to address point by point. All changes are highlighted in yellow. Thank you again for your help and support.

Comment 1: The introduction is too long and it is not focused, i.e. in lanes 74-80 the effects  of STAT3 inhibition are described, however these previous data  are not relevant for the study.

We have reduced the introduction from lines 30-94 to 30-89

Lines 74-80 have been modified to match the purpose as follows:

“Recent studies have shown that NDV reduced the expression and release of ICD markers in melanoma cells [20, 21]. Further investigation is required to establish whether oncolytic NDV successfully triggers ICD in specific malignancies.” (lines 73-76).

Comment 2: Authors describe the strain NDV Hitchener B1, but it is not clear whether this strain was used in the study

It is stated in Materials and Methods (line 99) that the virus used is Hitchner B1 as follows:

“The Hitchner B1 strain of Newcastle disease virus employed in this study, provided by Professor Zaher”

Comment 3: The authors have used throughout the study TCL, however it is not described the purpose of using TCL.

I agree. The purpose of TCL is mentioned in the Discussion section, lines 418-420, as follows.

“TCL was used in order to induce a more integrated immune response, as TCL is suitable for delivering a wide variety of antigens associated with MHC class I and II molecules [26].”

Comment 4: Authors have assessed the effects of NDV against human colon cell lines HCT16 and HT-29. However the effects have been evaluated in vivo against a mouse tumour model. I think authors should evaluate the effects of NDV also against mouse carcinoma colon cells.

This is done in Experiment 1, where mouse carcinoma colon cells were taken from xenograft mice and inoculated into mice lines 173-191, and the result was investigated lines 323-341.

Comment 5: Finally the discussion is too long and contains , again,  a description of the experiments performed,  instead of discussing the  results obtained. Authors must re-write d the discussion  ,  comparing the data obtained with previously published papers and clearly presenting  the limitations of the study as well the  relevance of their data.

Done. The discussion has been reduced from 416-531 to 402-490.

Reviewer 2 Report

Comments and Suggestions for Authors

The oncolytic effect of Newcastle disease virus (NDV) has been reported as early as the 1950s. Later, the oncolytic effect of NDV against various solid tumors was documented. However, the effect of NDV in human tumors remains inconclusive. The authors examined the oncolytic and immunomodulatory effects of NDV in two human colorectal cancer cell lines. This is a preclinical study utilizing in vitro and in vivo experiments with mouse models. The manuscript sheds light on the applicability of oncolytic NDV.

The manuscript is well-illustrated with 8 figures and is supported by 57 references. In my opinion, it can be accepted for publication after addressing the following minor points.

Minor points:

  • Consider presenting Figure 1 in black and white to potentially improve the impact of the photos.
  • Include measuring units for the X-axis in Figure 2.
  • Ensure that Figure 4A's appearance matches that of all other graphs for consistency.

Author Response

First, thank you for your time and effort in reviewing our manuscript and for your valuable remarks that helped make it the best it can be. We sincerely thank the reviewers for their valuable comments, which we will do our best to address point by point. All changes are highlighted in yellow. Thank you again for your help and support.

Comment 1: Consider presenting Figure 1 in black and white to potentially improve the impact of the photos.

Done. 

Comment 2: Include measuring units for the X-axis in Figure 2.

Done.

Comment 3: Ensure that Figure 4A's appearance matches that of all other graphs for consistency.

Done. 

Round 2

Reviewer 1 Report

Comments and Suggestions for Authors

Authors have improved the manuscripy as requested